# Dynamic Effects of Material Production and Environmental Sustainability on Economic Vitality Indicators: A Panel VAR Approach

**João Leitão** [1,2,3,4,*] and **Joaquim Ferreira** [2]

1   NECE-Research Center in Business Sciences, Faculty of Social and Human Sciences, University of Beira Interior, 6200-001 Covilhã, Portugal
2   NECE-Research Center in Business Sciences, University of Beira Interior, 6200-209 Covilhã, Portugal; joaquim.ferreira@ubi.pt
3   Centre of Management Studies of Instituto Superior Técnico (CEG-IST), University of Lisbon, 1649-004 Lisboa, Portugal
4   Instituto de Ciências Sociais (ICS), University of Lisbon, 1600-189 Lisboa, Portugal
*   Correspondence: jleitao@ubi.pt; Tel.: +351-275-319-853

**Abstract:** This study analyzes the relationships and dynamics between material production, foreign direct investment (FDI), economic activity, carbon productivity, the stock market, and green tech, both in a global and European context, using panel vector autoregressive methodology (PVAR). The empirical evidence obtained for the Global Group reveals four significant and positive unidirectional causality relationships, where aggregate material production is the prominent variable. For the EU-15 group, six significant causality relationships were detected, among them three negative and three positive unidirectional relationships. The stock markets shock reveals to be the most dominant variable, despite FDI standing out as causing the greatest shock effect. Nevertheless, in the European context, limited evidence of dematerialization is detected. Economic recessions show a generally negative effect, which contrasts with the economic Kitchin cycles, which reveal the effect of a generally positive relationship.

**Keywords:** economic activity; environmental sustainability; cycle; materials; panel models; sustainable finance

## 1. Introduction

The current scenario of global warming has led world institutions and political decision-makers to join various international discussion forums on mitigation strategies and the implementation of policies and measures to fight pollution and, generally, to adopt clean energy sources. However, these efforts have had a limited effect on the co-evolution of economic and population growth trajectories, resulting in an increasing demand for, and use of, natural resources and higher greenhouse gas emissions. This raises the question of the obligatory nature of sustainable economic development, aiming to implement a circular economy model as opposed to today's dominant model centered on fossil energy production and the exploitation of resources. In the same line of thought, the literature contains concepts and studies of reference that can illustrate better the still unexplored issue of the relationship between dematerialization and sustainable growth.

Highlighted first is the concept of dematerialization, characterized by decreasing use of material in the process of producing final products (Herman et al. 1990). It also serves to define a reduction in the intensity of raw material in economic activity (Bernardini and Galli 1993) or relative or absolute reduction of the amount of material and waste generated per production unit in the economy (Cleveland and Ruth 1998).

For there to be effective dematerialization of the economy, two processes stand out: the recycling process, which improves the product's quality and extends its useful life

(Herman et al. 1990); and the introduction of new technology, which consists of improving products' characteristics, leading to new products with less intensive use of materials (Tilton 1991; Bernardini and Galli 1993).

From a perspective of an harmonious relationship between growth and the environment, the literature applies the theoretical concept of the Environmental Kuznets Curve (EKC), or inverted U-shaped curve, according to which in a first stage of industrialization, the growth in domestic income accompanies increased environmental damage up to a certain threshold, with this diminishing a posteriori while income continues to grow (Grossman and Krueger 1991). In this view, the inverted U-shaped curve occurs because at the initial stage of economic growth, countries concentrate on increasing employment and income (Dasgupta et al. 2002; Dinda 2004), and at a second stage they focus on improving the quality of the environment (Dinda 2004). In turn, the first phase of the EKC may present a more stable configuration, as long as institutional policies to discourage the exploitation of natural resources are implemented (Panayotou 1993). It is noted, however, that such policies lead to faster adjustment of environmental quality in a phase of high incomes (Panayotou 1997).

The EKC hypothesis has been partially ratified through free trade between economies, in this way contributing to increased environmental damage, particularly in developing economies (Stern 1998). From another angle, based on an economic model concentrating on the industrial sector, developing economies tend to present lower levels of environmental damage than developed ones because the latter is based on sectors of great accumulation of physical and human capital (Grossman and Krueger 1991).

In this context, and despite observing a tendency towards greater investment in clean energy and ecological modernization allied to relocation and internationalization of industry (Aleluia and Leitão 2011), the environmental situation, in developed countries, continues to suffer because the coal industry benefits from low electricity prices, as opposed to increased prices for domestic electricity (Jänicke et al. 1997).

Therefore, with the main reasons of limiting the transfer of resources, implementing ecosystem cycles in economic cycles and respecting resource reproduction, the Circular Economy concept is promoted. This is based on a production-consumption system that aims to maximize the productive system following the linear form: Nature->Society->Natural Material->Energy Flows; through cycles of material and renewable sources of energy (Korhonen et al. 2018).

In the current context of globalization and economic integration, interdependences and spillover effects occur among markets. Given the information on the application of penalties and sanctions to listed companies whose behavior has an environmental impact, bearish behavior has been observed in markets (Muoghalu et al. 1990; Laplante and Lanoie 1994).

In this line of analysis, this study aims to analyze the behavioral effects and dynamics of shocks between material production, environmental sustainability and economic-financial variables, using the panel VAR (PVAR) methodology (Love and Zicchino 2006; Abrigo and Love 2016). Following the same previously referred authors, this advanced econometric methodology prevents endogeneity issues. In addition, it is not based on prior theory regarding variables' relationships, and provides the possibility of using two forecasting techniques, such as the orthogonalized impulse-response functions, and the forecast error decomposition variance, to gauge forecast effects on the system. Comparing to the VAR model, the current methodology in use, allows heterogeneity in panel estimation procedures.

The main contributions to the literature lie in analyzing dynamic feedback behavior or that of one-directional shocks originating in the variables representing material production, environmental sustainability and economic-financial indicators, including exogenous factors and the short-term economic cycles of Kitchin. The present empirical study presents a two-fold contribution for the literature on sustainable finance: (1) analyzing the still unexplored relationships between materials production, green growth, innovativeness,

and macroeconomic fundamentals, in order to deepen the knowledge on how to foster the production and financial strategies oriented to a green economy pathway; and (2) unveiling dematerialization paths, in terms of the relationship between sustainable/green growth and macroeconomic fundamentals, considering the short cycles of Kitchin.

The empirical study is structured as follows. It starts with a review of the theoretical and empirical literature on the relationships among material production, FDI, economic activity, environmental sustainability, stock markets, and environmental technologies, and developing the research hypotheses. Secondly, the econometric model and respective specification, are displayed. Then the results are presented, together with their discussion. Finally, the conclusions are presented, providing the main evidence and implications for policy-makers, the limitations of the study, and the guidelines for future research.

## 2. Theoretical Framework, Evidence and Hypothesis Development

The subjects of environmental degradation and climate change, resulting from the accelerated growth of industrial and economic activity in the 20th century and the beginning of this one, have gained prominence in the principal international forums for political, environmental and scientific discussion.

In terms of a theoretical paradigm, there is a certain convergence around the thesis that dematerialization, according to the EKC hypothesis, is only confirmed for low levels of growth (Ayres and Van Den Bergh 2005), which can arise from the fact that the costs of exploiting resources are greater than the wealth produced (Kemp-Benedict 2018). According to the same theoretical framework of reference, the spread of technology and technological progress do not influence the dematerialization process (Magee and Devezas 2017).

In this connection, given the empirical evidence obtained previously, the stylized fact stands out that the dematerialization process occurs above all in low income economies (Steinberger and Krausmann 2011; Shao et al. 2017) or in periods of economic recession (Shao et al. 2017), whereas in developed economies, growing material consumption is shown as a current process (Agnolucci et al. 2017). In addition, although material productivity tends to increase, dematerialization does not become particularly evident, given the substantial increase in the world population, leading to increased use of material consumption per capita (Krausmann et al. 2009).

In the empirical literature of reference, it is also worth highlighting the observation of that dematerialization process, according to the EKC hypothesis, in the context of developed or industrialized economies (Canas et al. 2003; Guzmán et al. 2005; Dong et al. 2017; Pothen and Welsch 2019).

In short, it is indicated that a lack of synchrony between levels of economic activity and material production/consumption has become more evident (Vehmas et al. 2007; Zhang et al. 2017), despite also finding a tendency towards increased dematerialization in fast-growing emerging economies, such as the case of China (Dai and Liu 2018). Therefore, the following hypothesis is considered:

**Hypothesis 1 (H1).** *The economic activity and material production denote a negative causality relationship.*

Considering the causal nexus established in the literature between economic activity and $CO_2$ emissions, diverging visions are found, based on different empirical approaches, which are worth reviewing in the framework this study belongs to. On one hand, there is an indication of decoupling between economic activity and $CO_2$ emissions (Wu et al. 2018; Chen et al. 2018; Dai and Liu 2018; Vo et al. 2019), and on the other hand, it underlines the lack of any statistically significant relationships regarding the causal nexus of reference (Cai et al. 2018) as well as a moderate positive effect between economic activity and $CO_2$ emissions (Kalaitzidakis et al. 2018). This leads to the following research hypothesis:

**Hypothesis 2 (H2).** *The economic activity and carbon productivity present a positive causality relationship.*

Bringing to this study the issue related to the integration of markets and their interdependences, it is considered necessary to study the behavior of those markets when faced with a shock of material production and $CO_2$ emissions. However, the literature focuses on commodity markets, on carbon energy markets and renewable energy markets and hedging strategies, and does not generally examine the relationship between stock markets, in relation to a variation in material production or between stock markets and variation in carbon levels.

It should be pointed out that some studies find no relationship between commodity markets and stock markets (Huang et al. 1996; Singhal and Ghosh 2016), despite finding a positive relationship with oil company stocks (Huang et al. 1996). Spillover effects are not observed between metal commodity markets and the stock market (Irandoust 2017).

From another perspective, the weak performance of stock markets has a positive effect on oil commodity prices (Jain and Biswal 2016), indicating a negative relationship between these two types of market. Indeed, pointed out as examples of better hedging strategies are investment in stock and oil commodity markets, in that a fall in prices causes increased volatility, leading to a significant asymmetrical effect between prices of the commodity and of the stock markets (Sadorsky 2014). In addition, commodity markets emerge as markets of monetary compensation, market instruments and substitute instruments, concerning investments based on a share portfolio (Batten et al. 2010).

Therefore, the following hypothesis is formulated:

**Hypothesis 3 (H3).** *The stock market and material production denote a negative causality relationship.*

Concerning the relationship between stock markets and carbon productivity, the empirical literature only contains studies on the relationships between the indices of shares of reference and those of the carbon market.

To optimize the value hoped for from an asset portfolio, a short position in the oil and carbon markets is suggested (European Union Allowances), as opposed to a long position in stock markets (Luo and Wu 2016). However, the relationship with the carbon market is found to be heterogeneous, in that stock market performance has a negative effect on the volatility of the carbon markets of the EUA and ERU (European Reduction Units), and has a positive impact on the volatility of the CER (Certificated Emission Reduction) market (Reckling 2016).

The carbon market has a positive influence on the shares of green energy companies, while having a negative impact on those of fossil fuel companies (da Silva et al. 2016), forming positive spillover effects of the volatility of the carbon market on the green energy share market (Dutta et al. 2018).

It is noted that development of the financial market has stimulated the demand for clean energy (Mamun et al. 2018), which means a negative relationship between financial markets and $CO_2$ emissions (Paramati et al. 2016; Paramati et al. 2017).

Furthermore, when a given company faces judicial actions, penalties, sanctions or any information regarding environmental degradation, it will be evaluated negatively by investors who will heavily penalize its shares on the market (Muoghalu et al. 1990; Laplante and Lanoie 1994). Therefore, considering the literature presented, the following hypothesis is presented:

**Hypothesis 4 (H4).** *The stock market and carbon productivity have a positive causality relationship.*

Concerning the relationship between Foreign Direct Investment (FDI) and $CO_2$ emissions, there is a notable shortage of previous empirical evidence. However, it can be confirmed that the relationship between FDI and $CO_2$ emissions is essentially positive (Lau et al. 2014; Seker et al. 2015), this being more evident in the long term (Paramati et al. 2016), which demonstrates that the economy's degree of openness leads to increased $CO_2$ emis-

sions. Analyzing the impact of adopting new processes or alternative forms of technology on $CO_2$ emissions, a positive association is also found (Paramati et al. 2017), confirming the importance of multinationals implementing efficient technology and processes.

Consequently, the following research hypothesis is considered:

**Hypothesis 5 (H5).** *The FDI and carbon productivity have a negative causality relationship.*

The literature on the relationships of interaction between financial markets and FDI concludes that: currency devaluation stimulates foreign investors to acquire domestic assets (Froot and Stein 1991); FDI contributes to the progress of macroeconomic fundamentals (Claessens et al. 2001), promoting financial markets' development (Agbloyor et al. 2013), which means agents have a greater appetite for local assets such as investors and funds (Boyer and Zheng 2009), contributing to increased share prices (Alfaro et al. 2004; Lizardo and Mollick 2009; Azman-Saini et al. 2010). From the above, the following hypothesis is formulated:

**Hypothesis 6 (H6).** *The FDI and stock market denote a positive causality relationship.*

Considering the causal nexus established in the literature between FDI and economic activity, the efficiency of the former is found to be greater than domestic investment, inasmuch as developing economies, especially, face restrictions in accessing finance in international markets (De Gregorio 1992). Indeed, the positive effect is greater in economies with strong policies on international trade (De Gregorio 1992), above all those directed towards exports (Balasubramanyam et al. 1996).

FDI is a driver of technology transfer, contributing to economic growth (Li and Liu 2005; Leitão and Baptista 2011; Makiela and Ouattara 2018), above all in economies with great capacity in terms of technology absorption and human capital (Li and Liu 2005). Added to this is the fact that economies with better indicators of financial development (Lee and Chang 2009; Iamsiraroj and Ulubaşoğlu 2015) and commercial openness (Iamsiraroj and Ulubaşoğlu 2015) are more able to attract FDI.

Moreover, capturing FDI promotes productivity spillovers, above all backward spillovers (e.g., linkages with domestic firms in different industries, such as upstream suppliers) (Javorcik 2004).

However, it is also true that FDI can have a negative influence on exporting economies where the primary sector dominates, signaling that FDI is negatively related to the abundance of resources (Herzer 2012). Considering the above, the following hypothesis is formulated:

**Hypothesis 7 (H7).** *The FDI and economic activity present a positive causality relationship.*

In carrying out this study, it is also necessary to consider the importance of economic activity in determining financial market behavior. Here, the causal nexus between the stock market and economic activity is characterized by a positive relationship (Schwert 1990; Choi et al. 1999).

Consequently, markets' behavior is considered an important predictive indicator of the behavior of economic activity (Choi et al. 1999; Hassapis and Kalyvitis 2002). Furthermore, financial development has a dominant role in determining the level of economic activity, especially by determining the level of liquidity, which is positively related to economies' contemporary and future behavior (Levine and Zervos 1998).

There is also previous evidence of a positive correlation between bilateral commercial relationships and the stock market (Tavares 2009). Thus, the following research hypothesis is raised:

**Hypothesis 8 (H8).** *The stock market and economic activity have a positive causality relationship.*

More recently, the technological innovation appears in the global policy agenda as a means of carbon mitigation and for the transition to a sustainable and green economy. However, although some empirical evidence shows that technological innovation becomes an important factor in carbon mitigation (Fernández et al. 2018), other studies identify the possibility of a rebound effect (Magee and Devezas 2017; Wang et al. 2019; Cheng et al. 2019) or from another angle, technological development does not decrease gas emission (Samargandi 2017; Mensah et al. 2018).

Considering the previous evidences, the following hypothesis is considered:

**Hypothesis 9 (H9).** *The carbon productivity and green technology have a negative causality relationship.*

### 3. Methodology

*3.1. Econometric Model*

This study aims to determine the relationships of causality and the effects of exogenous shocks on economic-financial indicators, resource productivity indicators and material production indicators, using a VAR model with panel data (PVAR) (Love and Zicchino 2006; Abrigo and Love 2016).

The mathematical formulation of the PVAR model is as follows:

$$Y_{i,t} = Y_{it-1}A_1 + Y_{it-2}A_2 + \ldots + Y_{it-p+1}A_{p-1} + Y_{it-p}A_p + X_{it}B + \mu_i + \epsilon_{it}$$
$$i \in \{1, 2, \ldots, N\}, \ t \in \{1, 2, \ldots, T_i\}$$
(1)

where: *i* corresponds to countries encompassed in the present study; *t* is the time horizon for each *i*; $Y_{it}$ is a *kx1* vector of endogenous variables; $X_{it}$ is a *lx1* vector of exogenous variables; $\mu_i$ is a *1xk* vector of individual fixed effects; and and $\epsilon_{it}$ is a *1xk* vector of idiosyncratic errors ($\epsilon_{it}$~ i.i.d.). The *kxk* matrices: $A_1, A_2, \ldots, A_p, A_{p-1}$; and the *lxk* matrix: *B*; represent the estimated parameters. Therefore, the PVAR model assumes that cross-sections hold same units in data generating process, which result in common parameters in matrixes: $A_1$, $A_2, \ldots, A_p, A_{p-1}$; and *B*; encompassing heterogeneity through panel-specific fixed effects (Holtz-Eakin et al. 1988; Abrigo and Love 2016).

Bearing in mind that $\mu_i$ is correlated with the lagged regressors, the use of OLS estimator can lead to bias of the coefficients (Nickell 1981; Abrigo and Love 2016). For reducing potential bias, the Helmert transformation procedure is performed (Arellano and Bover 1995). This leads to removal of the future means (i.e., the average of the set of future observations available for each unit of time, *per* country studied), thereby contributing to orthogonality between the dependent variables and the lagged regressors, as well as allowing their use as instrumental variables[1] and use of the GMM estimator.

To allow analysis of the forecast error variance decomposition (FEVD) and the simulated coefficients of the impulse-response functions (IRF), the stability condition of the estimated model should be validated, i.e., the modulus eigenvalues of a companion matrix A should be in an interval [0, 1] (Lütkepohl 2005). After check the stability condition of the estimated model, FEVD and IRF are used, as these tools can determine the dynamics of the endogenous variables in relation to exogenous shocks. These tools are expressed as follows:

$$\text{FEVD} \equiv \phi_i = \begin{cases} I_K, & i = 0 \\ \sum_{j=1}^{i} \phi_{t-j} A_j, & i = 1, 2, \ldots \end{cases}$$
(2)

$$\text{IRF} \equiv Y_{it+h} - E[Y_{it+h}] = \sum_{i=0}^{h-1} e_{i(t+h-i)} \phi_i$$
(3)

To do so, the orthogonal decomposition of Cholesky is performed, whereby the order of variables' entry is decided primarily by the greater degree of exogeneity of each of the variables of the model's selected specification.

---

1　The instrumental variables are specified according to the procedures proposed by Holtz-Eakin et al. (1988).

### 3.2. Data, Variables, and Specification of the Model

This study analyzes the response dynamics of economic, financial, production and resource sustainability indicators, in relation to an exogenous shock.

Therefore, annual unbalanced panel data are used, referring to the period 1990–2016 for 24 countries (Argentina, Australia, Austria, Belgium, Brazil, Canada, China, Denmark, Finland, France, Germany, Greece, Ireland, Italy, Japan, Mexico, Netherlands, Norway, Portugal, Russia, Spain, Sweden, United Kingdom and United States of America). The period of the sample is justified for two reasons: (i) limited access to data; and (ii) this being the longest period available (with annual frequency) to carry out this study. The data were gathered from the following databases: Investing.com; UNCTAD; OECD Statistics; British Geological Survey; and World Bank.

In the specification selected for the model, five endogenous variables are considered, namely: $MAT\_PR_{it}$, representing aggregate production (in tons) of the groups of material selected[2] in each country included; $FDI_{it}$, representing entry flows of Foreign Direct Investment (FDI), deflated by the GDP deflator, in each country included; $CO_2\_PR_{it}$, which is an incomplete proxy for the carbon productivity of each country; $GDP\text{-}PC_{it}$, which represents the national wealth, at constant prices, of each country; $SMKT_{it}$, representing the stock market indices of reference of each country; and $ENV\_TECH_{it}$, representing the amount of environmental-related technologies (e.g., patents).

Concerning the variables selected, in Table 1 presented below, the associated concepts, description, units, and statistical sources are displayed.

**Table 1.** Variables selected.

| Variables | Associated Concepts | Description | Units | Statistical Sources |
|---|---|---|---|---|
| Material Production (MAT_PR) | Aggregate Material Production | Production of minerals commodities | Tons | British Geological Survey |
| Foreign Direct Investment (FDI) | Foreign Direct Investment or FDI | Inward and outward flows and stock | US Millions deflated by GDP deflator | UNCTAD |
| Gross Domestic Product (GDP_PC) | Economic Activity | Total of GDP per capita | US Millions in constant prices | UNCTAD |
| $CO_2$ productivity (CO$_2$_PR) | Carbon productivity | GDP per units of energy-related $CO_2$ emissions | US dollar per kilogram | OECD |
| Stock Markets (SMKT) | Stock Markets | Major domestic stock markets indexes | Index points | investing.com |
| Environment-related technologies (ENV_TECH) | Green Tech | Patents related with environmental management, water adaptation and climate change mitigation | Units of patents | OECD |

Source: Own elaboration.

The next step was logarithmic transformation of the series, in order to ensure greater convergence of the coefficients estimated and contribute to better adjustment of the model.

Aiming for a subsequent comparative analysis, the study's methodological design considers the possibility of determining the response dynamics in two groups of countries. The first group corresponds to all twenty-four countries in the sample (Global Group). The second corresponds to the 15 European Union countries[3] (EU-15 Group). For each group,

---

[2] Given the limited access to data, these were collected referring to the group of metals and the group of minerals. Concerning the group of metals, the materials included in the study variable are: aluminium, steel, cadmium, bismuth, lead, cobalt, copper, tin, iron, pig iron, lithium, magnesium, manganese, nickel, gold, silver, platinum and its derivatives and zinc. The group of minerals includes the following materials: asbestos, alumina, barite, feldspar, rock phosphate, gypsum, graphite, mica, salt and zirconia.

[3] For the European Union, only Luxembourg was not included due to the unavailability of data.

three dummy variables are considered, aiming to capture the main crises originated from emerging markets ($D^{Crises\ EM}$), and developed markets ($D^{Crises\ DM}$)$D^{Crises\ EM}D^{Crises\ DM4}$, covering two economic cycles of Kitchin[5], and another dummy variable that characterizes recession periods in the larger economies in each group $\left(D^{Global}; D^{EU}\right)D^{Global}D^{EU6}$.

The selected specification of the model is as follows:

$$Y_{it} = A_0 + A_1 Y_{t-p} + D^{Crises\ EM} + D^{Crises\ DM} + D^{Global/EU} + \mu_i + \epsilon_{it}, \epsilon_{it} \sim \text{i.i.d.}$$
$$i \in \{1, 2, \ldots, 24\}^{GLOBAL}, t \in \{1990, 1991, \ldots, 2016\}^{GLOBAL} \qquad (4)$$
$$i \in \{1, 2, \ldots, 14\}^{EU\text{-}15}, t \in \{1990, 1991, \ldots, 2016\}^{EU\text{-}15}$$

where, $Y_{it} \equiv \{MAT_{PRit}\ FDI_{it}\ GDP_{PCit}\ CO2_{PRit}\ SMKT_{it}\ ENV\_TECH_{it}\}$.

## 4. Results and Discussion

### 4.1. Empirical Evidence

The results obtained from estimating the PVAR model and evidence from the dynamic analysis are now presented, using three tools for testing and forecasting: Granger causality, FEVD, and IRF.

Before estimating the model, diagnostic tests were performed to ensure no misspecification. To do so, the cross-sectional dependence was verified (Pesaran 2015), which led to applying the unit root test CIPS (Pesaran 2007), proceeding to differentiation of the variables, in order to ensure they became stationary or integrated of order zero, that is, $I(0)$[7]. To determine the optimal number of lags and moments, the Andrews and Lu (2001) test was applied. It admits one optimal lag, considering from two until five lags (in both groups) concerning instrumental variables. The optimal number of lags was selected against application of the criterion that minimize the *J*-statistic of Hansen (1982).

In order to compare the two groups studied, i.e., Global Group and EU-15 Group, and after performing the introductory tests for estimation of the PVAR[8] model, the coefficients estimated were obtained (cf. Tables 2 and 3).

In Table 2 presented below, referring to the Global Group, the values obtained for the *J*-statistic of Hansen (1982) determine that the null hypothesis is not rejected, thereby ratifying the validity of the instruments used in estimating the model.

In model 1, and the dependent variable being $MAT\_PR_t$; the variables with greatest statistical significance are $MAT\_PR_{t-1}$ and $GDP\_PC_{t-1}$, with a positive effect and a negative effect, respectively, at the 5% level. The Dummy Crises DM shows a positive and statistically significant effect, at the 5% level, whereas the Dummy Global affects negatively and significantly, at the 1% significance level.

In model 2, with the dependent variable: $FDI_t$; the variables $FDI_{t-1}$ and $SMKT_{t-1}$ are the most predominant, exhibiting a positive and significant effect, at the 1% and 5% significance level, respectively. The $MAT\_PR_{t-1}$ and $CO_2\_PR_{t-1}$ denote a positive and negative effect on the behavior of $FDI_t$, respectively, with associated statistical significance

---

[4]  The dummy variable $D^{Crises\ EM}$ has the value of 1 in the annual periods of 1991, 1994, 1995, 1997–2000, 2002, and the value of zero in the remaining periods. The periods under analysis correspond to different international crises, such as: the oil crisis (1991); the Mexican economic crisis (1994/1995); the Asian monetary crisis (1997); the Russian monetary crisis (1998); the Brazilian monetary crisis (1999); the Argentinian economic crisis (1999–2000); and the South American economic crisis (2002). The dummy variable $D^{Crises\ EM}$ equals to 1 in the annual periods of 2001 and 2007–2010, and 0 in other periods. These periods correspond to the dotcom bubble (2001), the subprime crisis (2007–2008) and the European debt crisis (2009–2010).

[5]  The Kitchin cycles are classified as short-term cycles, i.e., cycles lasting 4 years. Therefore, the Kitchin cycles found in the period of analysis correspond to the periods 1997–2000 and 2007–2010.

[6]  The dummy variable $D^{Global}$ represents economic recession in the USA and People's Republic of China, having the value of 1 in the annual periods of 2008 and 2009 and the value of zero in the other periods. The dummy variable $D^{EU}$ corresponds to economic recession in Germany, France and the United Kingdom, having the value of 1 in the annual periods of 1991–1992, 2002–2003 and 2008–2009 and the value of zero in the other periods analysed.

[7]  In the Global Group, the FDI, SMKT and ENV_TECH variables appear as stationary at levels, whereas in the EU-15 Group only FDI is stationary, at levels.

[8]  The tables of the tests applied can be obtained upon request to the authors.

of 10%. Therefore, the Dummy Crises EM is found to have a positive and significant effect on FDI$_t$, at the 1% significance level, whereas the Dummy Global affects negatively the FDI$_t$, at the 10% level.

In model 3, with the dependent variable: GDP_PC$_t$; there are positive effects of MAT_PR$_{t-1}$ and GDP_PC$_{t-1}$, at the 1% significance level. In turn, both the Dummy Crises and Dummy Global, there is mixed evidence, detecting a positive and negative effect, respectively, at a 1% level of significance.

In model 4, considering as dependent variable: CO$_2$_PR$_t$; GDP_PC$_{t-1}$ and ENV_TECH$_{t-1}$ are found to have a positive, and significant effect, at the 1% significance level, while for the variable CO$_2$_PR$_{t-1}$ negative and significant effects are found, at the 1% significance level either.

In model 5, with the dependent variable: SMKT$_t$; SMKT$_{t-1}$ and CO$_2$_PR$_{t-1}$ are found to have a positive and significant effect, at the 1% and 5% significance level, respectively. The Dummy Global presents, within the group of dummy variables, as the predominant insofar as affects negatively and significantly the FDI$_t$, at the 1% statistical significance level.

In model 6, with the dependent variable: ENV_TECH$_t$; the variables ENV_TECH$_{t-1}$ and MAT_PR$_{t-1}$ perform as the predominant ones, insofar as affect positively the FDI, at the 1% and 5% significance level. Regarding the Dummy Crises EM and Dummy Crises DM evidence positive and significant effect, at the 1% and 5% significance level, respectively. Unlike, the Dummy Global affects negatively at the 10% significance level.

**Table 2.** The Global Group PVAR estimators.

| Models | (1) | (2) | (3) | (4) | (5) | (6) |
|---|---|---|---|---|---|---|
| Dependent Variable | MAT_PR | FDI | GDP_PC | CO$_2$_PR | SMKT | ENV_TECH |
| MAT_PR | 0.4120 ** | 1.9798 * | 0.0719 *** | −0.0941 | 0.3907 | 1.0105 ** |
| | [2.5700] | [1.9000] | [3.0400] | [−1.2700] | [1.3500] | [2.3700] |
| FDI | −0.0136 | 0.8452 *** | −0.0030 | 0.0038 | 0.0235 | −0.0048 |
| | [−1.3000] | [8.8300] | [−1.5500] | [0.8000] | [0.7900] | [−0.1700] |
| GDP_PC | −1.0465 ** | 0.0254 | 0.3489 *** | 0.8836 *** | −1.1130 | −0.6140 |
| | [−1.9700] | [0.0100] | [3.4100] | [3.3700] | [−0.7500] | [−0.4000] |
| CO$_2$_PR | 0.5478 | −5.2331 * | −0.0535 | −0.5501 *** | 1.7834 ** | 0.5847 |
| | [1.4300] | [−1.7200] | [−0.7800] | [−3.1900] | [2.3000] | [0.6500] |
| SMKT | −0.0147 | 0.1931 ** | −0.0033 | −0.0058 | 0.8226 *** | 0.0146 |
| | [−1.3500] | [2.0900] | [−1.6200] | [−1.1800] | [23.0100] | [0.4500] |
| ENV_TECH | −0.0231 | 0.0671 | 0.0030 | 0.0209 *** | 0.0335 | 0.9463 *** |
| | [−1.6400] | [0.5400] | [0.9700] | [2.8400] | [0.7400] | [20.0700] |
| Dummy Crises EM | −0.0069 | 0.2746 ** | 0.0110 *** | 0.0080 | −0.0589 | 0.0054 |
| | [−0.4700] | [1.6100] | [4.6200] | [1.2600] | [−1.6400] | [0.1300] |
| Dummy Crises DM | 0.0560 ** | 0.0575 | 0.0189 *** | −0.0010 | −0.0146 | 0.1384 ** |
| | [2.3300] | [0.3200] | [5.3600] | [−0.1100] | [−0.3300] | [2.4800] |
| Dummy Global | −0.1571 *** | −0.3331 * | −0.0514 *** | −0.0037 | −0.3125 *** | −0.1140 * |
| | [−4.7100] | [−1.9600] | [−9.3300] | [−0.3800] | [−4.7100] | [−1.9100] |

Legend: Test of over identifying restriction: Hansen's J Chi$^2$ (108) = 121.8580 (*p* = 0.171). Notes: ***, **, * indicate significance at 1%, 5% and 10%, respectively. Z-statistics are in square brackets. Source: Own elaboration.

In the comparative analysis, in Table 3, presented below, referring to the EU-15 Group, it is highlighted that given the values obtained for the *J*-statistic of Hansen (1982), the null hypothesis is not rejected, thereby ratifying the validity of the instruments used in estimating the model.

Taking as a reference the results of model 1's estimation, with the dependent variable being MAT_PR$_t$; both GDP_PC$_{t-1}$ and SMKT$_{t-1}$ show negative and positive significant effects, respectively, at the 1% significance level. In turn, concerning exogenous variables, only the Dummy Crises reveals a positively significant effect, at a 1% level.

In model 2, considering as dependent variable: $FDI_t$; the lagged variable, that is, the $FDI_{t-1}$, denotes statistical significance, at a 1% level, with positive effects. In turn, the $SMKT_{t-1}$ reveals a negative and significant effect, at a 10% level. The dummy variables present statistical significance at the 1% level, albeit it should be noted that the Dummy Crises EM has a positive effect, contrasting with the negative effect of the Dummy EU.

In model 3, considering as dependent variable: $GDP\_PC_t$; the lagged variable, that is, the $GDP\_PC_{t-1}$, reveals to be the unique endogenous variable that affects significantly and positively the behavior of $GDP\_PC_t$, at the 1% significance level. The dummy variables, mainly, the Dummy Crises EM and the Dummy EU present the opposite effects, at the 1% significance level.

For model 4, with the dependent variable: $CO_2\_PR_t$; on the one hand, variables of $MAT\_PR_{t-1}$, $FDI_{t-1}$, and $ENV\_TECH_{t-1}$, have a positive and statistically significant effect at 5% level. On the other hand, the variable $CO_2\_PR_{t-1}$ have a negative and statistically significant effect, at 5% level.

In model 5, with the dependent variable: $SMKT_t$; $CO_2\_PR_{t-1}$ and $SMKT_{t-1}$ produce positive effects at the 1% and 10% significance level, respectively, while in the $GDP\_PC_{t-1}$ is found opposite effects at the 5% significance level. In the case of the dummy variables, only Dummy Crises DM affects in a negative and significantly way the $SMKT_t$, at 1% level.

In model 6, with the dependent variable: $ENV\_TECH_t$; the $MAT\_PR_{t-1}$ is the unique endogenous variable that impacts significantly $ENV\_TECH_t$, with a negative sign at the 10% significance level. Concerning dummy variables, only Dummy Crises EM affects positively and significantly, however, at the 10% significance level.

**Table 3.** The EU-15 Group PVAR estimators.

| Models | (1) | (2) | (3) | (4) | (5) | (6) |
|---|---|---|---|---|---|---|
| Dependent Variable | MAT_PR | FDI | GDP_PC | CO$_2$_PR | SMKT | ENV_TECH |
| MAT_PR | 0.0980 | 0.9915 | 0.0075 | 0.1032 ** | −0.2974 | 0.4450 * |
| | [0.7700] | [1.0300] | [0.3200] | [2.4100] | [−0.9400] | [1.8600] |
| FDI | 0.0129 | 0.9471 *** | −0.0049 | 0.0165 ** | −0.0058 | 0.0396 |
| | [0.5900] | [5.9600] | [−1.3600] | [2.0600] | [−0.1000] | [1.1500] |
| GDP_PC | −2.3522 *** | −1.6165 | 0.2052 *** | 0.1862 | −2.4965 ** | −0.3280 |
| | [−4.7900] | [−0.4900] | [2.7900] | [1.1400] | [−1.9700] | [−0.3900] |
| CO$_2$_PR | 0.3538 | −1.8769 | 0.0410 | −0.2540 *** | 2.1104 *** | 0.2990 |
| | [1.4800] | [−1.0600] | [0.9300] | [−2.9000] | [3.0000] | [0.7600] |
| SMKT | 0.2345 *** | −0.7736 * | 0.0153 | 0.0249 | 0.3182 * | 0.0170 |
| | [3.5100] | [−1.6900] | [1.4800] | [1.0300] | [1.8600] | [0.1700] |
| ENV_TECH | −0.1162 | −0.2358 | −0.0172 | 0.0509 ** | 0.0798 | 0.0259 |
| | [−1.4600] | [−0.5200] | [−1.4400] | [2.3700] | [0.4900] | [0.1800] |
| Dummy Crises EM | 0.0383 ** | 0.4088 *** | 0.0135 *** | −0.0075 | −0.0205 | 0.0475 * |
| | [2.0800] | [3.1600] | [4.7500] | [−1.1900] | [−0.4600] | [1.7600] |
| Dummy Crises DM | −0.0195 | 0.0029 | 0.0039 | −0.0103 | −0.2392 *** | 0.0553 |
| | [−0.8000] | [0.0100] | [1.0000] | [−1.1900] | [−4.2400] | [1.6200] |
| Dummy EU | 0.0247 | −0.6449 *** | −0.0190 *** | −0.0114 | −0.0030 | 0.0339 |
| | [0.8800] | [−3.1200] | [−4.4800] | [−1.1600] | [−0.0300] | [0.7400] |

Legend: Test of over identifying restriction: Hansen's J Chi$^2$ (175) = 117.91124 ($p$ = 0.242). Notes: ***, **, * indicate significance at 1%, 5% and 10%, respectively. Z-statistics are in square brackets. Source: Own elaboration.

The graphic representation presented below in Figure 1 reveals that the modulus of the eigenvalues of the companion matrix is within the unit circle, concluding therefore that the PVAR model satisfies the condition of stability, demonstrating that it is invertible and representing an infinite-order vector moving average, allowing estimation of the forecast error variance decomposition and coefficients of the impulse-response functions.

Global Group

EU-15 Group

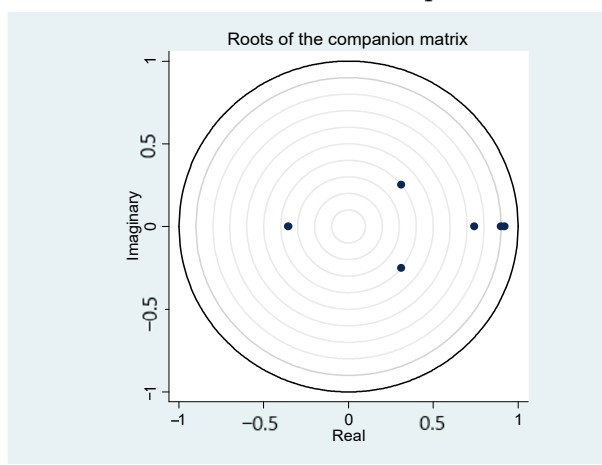

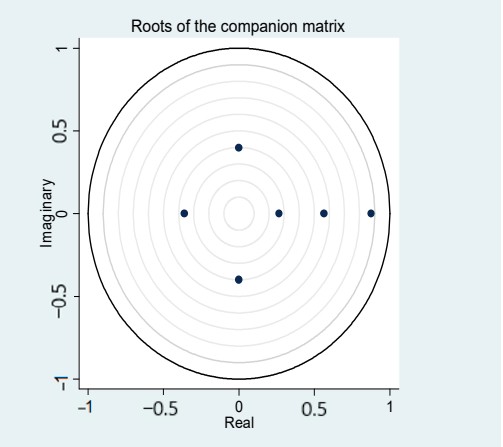

Source: Own elaboration.

**Figure 1.** Eigenvalue stability condition.

As proposed by Abrigo and Love (2016), to study relationships of causality and amplitudes[9], dynamic analysis of the PVAR model is based on applying either the forecast error variance decomposition technique or the technique of impulse-response functions orthogonalized by Cholesky decomposition, with 200 Monte Carlo replications.

From the evidence obtained from that dynamic analysis, firstly for the Global Group (cf., Table 4), four causality relationships with a significant impact are found. Therefore, firstly, a shock arising from aggregate material production is seen to have a positive and significant impact on FDI, because, over the forecasting periods, FDI is explained around 8.4% through the shock in aggregate material production. Secondly, gross domestic product per capita responds positively and significantly to a shock in aggregate material production, ranging between 29% and 28%, over the forecast period. Thirdly, during the forecasting period, an economic activity shock leads to a positive and significant effect on carbon productivity at around 9%. Fourthly, there is a positive effect from aggregate material production shock on the green tech of 21.5%, after eight periods.

In overall calculation of the results of the analysis of causality and amplitude relationships, concerning the first group of countries, that is, the Global Group, the shock from aggregate material production stands out as the prominent one, revealing three causality relationships.

As for the EU-15 Group (cf. Table 5) is concerned, a greater number of significant relationships are found, i.e., six causality relationships with significant effects. Consequently, aggregate material production is now found to respond significantly and negatively to an economic activity shock between 10% and 8%, whose significant effect is not identified in Global Group results. In addition, bearing in mind a shock from the stock market, aggregate material production responds positively and significantly between 19% and 17%, during the forecasting period. In turn, a shock from stock markets has a negative and significant effect on FDI, between 5% and 7%, during the forecasting period, contrasting the outcomes from Global Group. Therefore, the aggregate material production shock produces a positive and significant effect on carbon productivity of between 9% and 7%, whilst a negative and significant shock from FDI presents an effect between 20% and 37%, during the forecasting period. Shocks from carbon productivity contribute positively and significantly to stock markets behavior between 9% and 8%. It is also worth pointing out that, for the EU-15 Group, the stock market shock becomes the predominant shock, despite the FDI inducing the higher effect amplitude.

---

9   The PVAR Granger causality Wald test results, for the sake of brevity, can be obtained on request from the authors.

**Table 4.** Analysis of causality and amplitude relationships in the Global Group.

| Equantion Variable | Excluded Variable | Dynamic Analysis | 4 Years | 8 Years | 10 Years | Sign |
|---|---|---|---|---|---|---|
| MAT_PR | | | | | | |
| | GDP_PC | FEVD | 0.0312 | 0.0307 | 0.0306 | - |
| | | COIRF | −0.0289 | −0.0241 | −0.0228 | |
| FDI | | | | | | |
| | MAT_PR | FEVD | 0.0842 | 0.0834 | 0.0825 | ⊥ |
| | | COIRF | 0.8485 | 1.1720 | 1.2737 | |
| | CO₂_PR | FEVD | 0.0289 | 0.0238 | 0.0229 | - |
| | | COIRF | −0.3221 | −0.3905 | −0.4071 | |
| | SMKT | FEVD | 0.0122 | 0.0318 | 0.0390 | + |
| | | COIRF | 0.3957 | 0.8743 | 1.0787 | |
| GDP_PC | | | | | | |
| | MAT_PR | FEVD | 0.2924 | 0.2821 | 0.2799 | ⊥ |
| | | COIRF | 0.0263 | 0.0219 | 0.0206 | |
| CO₂_PR | | | | | | |
| | GDP_PC | FEVD | 0.0903 | 0.0894 | 0.0891 | ⊥ |
| | | COIRF | 0.0226 | 0.0222 | 0.0222 | |
| | ENV_TECH | FEVD | 0.0153 | 0.0229 | 0.0251 | + |
| | | COIRF | 0.0152 | 0.0248 | 0.0285 | |
| SMKT | | | | | | |
| | CO₂_PR | FEVD | 0.0192 | 0.0165 | 0.0160 | + |
| | | COIRF | 0.0838 | 0.1271 | 0.1410 | |
| ENV_TECH | | | | | | |
| | MAT_PR | FEVD | 0.1988 | 0.2155 | 0.2152 | ⊥ |
| | | COIRF | 0.5593 | 0.9260 | 1.0637 | |

Legend: FEVD—Forecast error variance decomposition; IRF—Cumulative Orthogonalized Impulse-Response Function. The causality sign is obtained from the accumulated value of the 10 periods' coefficients because from that period coefficients reach the necessary stability (Goux 1996). The direction of causality analyzed presents a significant impact, i.e., over 5% after eight periods (Goux 1996). Source: Own elaboration.

**Table 5.** Analysis of causality and amplitude relationships in the EU-15 Group.

| Equation Variable | Excluded Variable | Dynamic Analysis | 4 Years | 8 Years | 10 Years | Sign |
|---|---|---|---|---|---|---|
| MAT_PR | | | | | | |
| | GDP_PC | FEVD | 0.1047 | 0.0852 | 0.081 | - |
| | | COIRF | −0.0722 | −0.0080 | −0.0816 | |
| | SMKT | FEVD | 0.1940 | 0.1702 | 0.1672 | ⊥ |
| | | COIRF | 0.0657 | 0.0265 | 0.010 | |
| FDI | | | | | | |
| | SMKT | FEVD | 0.0485 | 0.0691 | 0.0728 | - |
| | | COIRF | −1.0312 | −1.8847 | −2.1822 | |
| CO₂_PR | | | | | | |
| | MAT_PR | FEVD | 0.0855 | 0.0715 | 0.0685 | ⊥ |
| | | COIRF | 0.0074 | 0.0149 | 0.0175 | |
| | FDI | FEVD | 0.2088 | 0.3415 | 0.3666 | - |
| | | COIRF | 0.0745 | 0.1285 | 0.1474 | |
| | ENV_TECH | FEVD | 0.0411 | 0.0347 | 0.0338 | - |
| | | COIRF | 0.0058 | −0.0008 | −0.0036 | |
| SMKT | | | | | | |
| | GDP_PC | FEVD | 0.0181 | 0.0176 | 0.0172 | - |
| | | COIRF | −0.0621 | −0.0759 | −0.0783 | |
| | CO₂_PR | FEVD | 0.0913 | 0.0847 | 0.0832 | ⊥ |
| | | COIRF | 0.1068 | 0.0800 | 0.0675 | |
| ENV_TECH | | | | | | |
| | MAT_PR | FEVD | 0.0311 | 0.0286 | 0.0280 | + |
| | | COIRF | 0.0633 | 0.0899 | 0.0992 | |

Legend: FEVD—forecast error variance decomposition; IRF—cumulative orthogonalized impulse response function. The causality sign is obtained from the accumulated value of the 10 periods coefficients because from that period coefficients reach the necessary stability (Goux 1996). The direction of causality analyzed presents a significant impact, i.e., over 5% after eight periods (Goux 1996). Source: Own elaboration.

### 4.2. Robustness of the Model

To determine the robustness of the model estimated, a change is introduced in the entry of endogenous variables in the Cholesky decomposition of the forecast error variance. The vector of endogenous variables introduced is described as follows:

$$Z_{it} \equiv \{SMKT_{it} \; ENV\_TECH_{it} \; FDI_{it} \; CO2\_PR_{it} \; MAT\_PR_{it} \; GDP\_PC_{it}\} \tag{5}$$

Concerning the Global Group (cf. Table 6), in the results obtained through the robustness test, no change in the typology of the sign is observed, and there is no change to the significance of the relationships, taking the estimators obtained for the benchmark model as a reference.

**Table 6.** Robustness test for the Global Group.

| Equation Variable | Excluded Variable | Dynamic Analysis | 4 Years | 8 Years | 10 Years | Sign |
|---|---|---|---|---|---|---|
| MAT_PR | | | | | | |
| | GDP_PC | FEVD | 0.0365 | 0.0369 | 0.0370 | - |
| | | COIRF | −0.0272 | −0.0182 | −0.0154 | |
| FDI | | | | | | |
| | MAT_PR | FEVD | 0.0724 | 0.0735 | 0.0730 | ⊥ |
| | | COIRF | 0.8137 | 1.1337 | 1.2359 | |
| | CO$_2$_PR | FEVD | 0.0474 | 0.0418 | 0.0406 | - |
| | | COIRF | −0.5433 | −0.7033 | −0.7446 | |
| | SMKT | FEVD | 0.0083 | 0.0231 | 0.0292 | + |
| | | COIRF | 0.2945 | 0.7175 | 0.9048 | |
| GDP_PC | | | | | | |
| | MAT_PR | FEVD | 0.2862 | 0.2757 | 0.2735 | ' |
| | | COIRF | 0.0265 | 0.0223 | 0.0209 | |
| CO$_2$_PR | | | | | | |
| | GDP_PC | FEVD | 0.1096 | 0.1087 | 0.1084 | ⊥ |
| | | COIRF | 0.0166 | 0.0156 | 0.0155 | |
| | ENV_TECH | FEVD | 0.0355 | 0.0426 | 0.0447 | + |
| | | COIRF | 0.0159 | 0.0254 | 0.0291 | |
| SMKT | | | | | | |
| | CO$_2$_PR | FEVD | 0.0261 | 0.0218 | 0.0208 | + |
| | | COIRF | 0.1404 | 0.1790 | 0.1882 | |
| ENV_TECH | | | | | | |
| | MAT_PR | FEVD | 0.1397 | 0.1531 | 0.1527 | ⊥ |
| | | COIRF | 0.4489 | 0.7584 | 0.8735 | |

Legend: FEVD—forecast error variance decomposition; IRF—cumulative orthogonalized impulse Response function. The causality sign is obtained from the accumulated value of the 10 periods coefficients because from that period coefficients reach the necessary stability (Goux 1996). The direction of causality analyzed presents a significant impact, i.e., over 5% after eight periods (Goux 1996). Source: Own elaboration.

Regarding the EU-15 Group (cf. Table 7), it is observed a reduction in the total number of significant causality relationships, from six to five relationships, comparing with the benchmark model. On the one hand, it reveals that the relationship in which carbon productivity responds an aggregate material production shock is not significant, contrasting with the results of the benchmark model. On the other hand, the typology of the signal in some relationships is changed compared to the benchmark model. Thus, the significant relationships in which the aggregate material production response to a shock from stock markets, as well as the carbon productivity response to a shock from FDI, reveal a negative relationship, in opposite to results from the benchmark model. Nevertheless, it should be noted that a robustness check based in a Cholesky Decomposition (with a lower or upper triangular matrix), changing the variables ordering, affects, somehow, the amplitude of shocks and signal typology either. Adding to the previous, it can be observed that significant relationships converge with PVAR estimates. Hence, as it is verified a

switch on signal typology in two significant relationships and on the amplitude of shocks in one significant relationship, it can be argued that the model, in the Global Group and, above all, in the EU-15 Group, shows statistical robustness.

**Table 7.** Robustness test for the EU-15 Group.

| Equation Variable | Excluded Variable | Dynamic Analysis | 4 Years | 8 Years | 10 Years | Sign |
|---|---|---|---|---|---|---|
| MAT_PR | | | | | | |
| | GDP_PC | FEVD | 0.1197 | 0.0971 | 0.0920 | - |
| | | COIRF | −0.0748 | −0.0822 | −0.0835 | |
| | SMKT | FEVD | 0.1769 | 0.1677 | 0.1683 | - |
| | | COIRF | 0.0536 | 0.0025 | −0.0181 | |
| FDI | | | | | | |
| | SMKT | FEVD | 0.0911 | 0.1194 | 0.1244 | - |
| | | COIRF | −1.4795 | −2.5512 | −2.9221 | |
| CO$_2$_PR | | | | | | |
| | MAT_PR | FEVD | 0.0340 | 0.0281 | 0.0268 | + |
| | | COIRF | 0.0122 | 0.0161 | 0.0174 | |
| | FDI | FEVD | 0.2106 | 0.3342 | 0.3573 | ⊥ |
| | | COIRF | 0.0741 | 0.1267 | 0.1451 | |
| | ENV_TECH | FEVD | 0.0459 | 0.0369 | 0.0351 | + |
| | | COIRF | 0.0153 | 0.0132 | 0.0120 | |
| SMKT | | | | | | |
| | GDP_PC | FEVD | 0.0202 | 0.0195 | 0.0190 | - |
| | | COIRF | −0.0716 | −0.0845 | −0.0866 | |
| | CO$_2$_PR | FEVD | 0.1093 | 0.1014 | 0.0996 | ' |
| | | COIRF | 0.1090 | 0.0800 | 0.0667 | |
| ENV_TECH | | | | | | |
| | MAT_PR | FEVD | 0.0297 | 0.0254 | 0.0245 | + |
| | | COIRF | 0.0472 | 0.0612 | 0.0659 | |

Legend: FEVD—forecast error variance decomposition; IRF—cumulative orthogonalized impulse Response function. The causality sign is obtained from the accumulated value of the 10 periods coefficients because from that period coefficients reach the necessary stability (Goux 1996). The direction of causality analyzed presents a significant impact, i.e., over 5% after eight periods (Goux 1996). Source: Own elaboration.

*4.3. Discussion*

Economies are based on a fossil energy model, with a notable correlation between material consumption and economic activity (Steinberger and Krausmann 2011), which in turn contributes to even greater stimulation of the socioeconomic metabolism (Krausmann et al. 2009), with the driving levers of low energy and material prices (Agnolucci et al. 2017). Nevertheless, the results obtained, in the European context, display a negative effect of the economic activity on material production, in the EU-15 Group, which denote that material production decreases as it ramps the development and income state up of an economy, according to with EKC (Canas et al. 2003) and through environmental policies implemented (Vehmas et al. 2007). Thus, H1 is rejected for Global Group but it is not rejected for the EU-15 Group.

Considering the factual evidence found in some empirical literature, according to which emerging economies are presented as an important factor in the major decoupling of developed countries, due to industries' relocation (Wu et al. 2018) or policies of environmental regulation, there is a notable agreement between the evidence obtained here and the above arguments. Therefore, the rate of population growth becomes the main driver of material and energy consumption (Chen et al. 2018), as well as the growth of gross domestic product reveals to be one of the driving forces of CO$_2$ emissions (Vo et al. 2019). It turns out that besides government incentives for clean energy consumption, the onset of financial crises as recession cycles can imply an increase in carbon productivity. Thus, H2 is not rejected for the Global Group, but it is rejected for the EU-15 Group.

In turn, referring to previous evidence that precious metal markets emerge as substitutes for stock markets (Batten et al. 2010; Jain and Biswal 2016) or showing a structure of hedging of greater risk (Sadorsky 2014), the results revealed here point to a set of contradictory evidence. Consequently, the evidence obtained may indicate a certain operational efficiency of equity markets, not admitting the adoption of arbitrage practices (Irandoust 2017), which indicates rejection of H3, for both Groups.

Concerning the stock markets of European economies, on one hand these emerge as important drivers of green/renewable energy consumption (Paramati et al. 2016), through listed companies' absorption of green energy technology (Paramati et al. 2017). On the other hand, increased volatility of carbon markets means an unfavorable shock for investment (Reckling 2016), which can contribute to environmental sustainability having a negative impact on European markets. In this context, H4 is rejected both for the Global Group, and for the relationship between carbon productivity and the stock market, for the EU-15 Group. However, H4 is not rejected for the stock market-carbon productivity causality relationship between the stock market and carbon productivity, for the EU-15 Group.

The empirical evidence regarding the FDI and carbon productivity relationship reveals that the trade liberalization increases $CO_2$ emissions (Lau et al. 2014), even taking into consideration that the FDI ensured by multinational companies, through their more efficient and clean energy technology, leads to reduced $CO_2$ emissions (Paramati et al. 2016). Notwithstanding, financial crises and expensive clean-related technologies costs arise as constraints to achieve carbon mitigation. Therefore, H5 is rejected for Global Group but it is not rejected for EU-15 Group.

The FDI is a key-driver of investment dynamics especially in developed financial systems (Azman-Saini et al. 2010), which allows greater freedom in capital transactions (Agbloyor et al. 2013) and the adoption of diversification strategies by investors (Lizardo and Mollick 2009). Furthermore, the internationalization of these markets contributes to increased FDI, through greater market capitalization (Claessens et al. 2001). Nevertheless, the empirical evidence of this empirical study only depicts a negative relationship between FDI and stock markets in the European context. Such evidence arises from stock markets volatility and uncertainty fostered by subprime and European sovereign debt crises. Therefore, H6 is rejected for both Groups.

In turn, FDI and economic activity evidence no nexus of causality, justified by the great uncertainty in the European and international economic and political situation, which discourages investment (Herzer 2012). This indicates rejection of H7 for both Groups.

The high level of financial and economic integration, as well as existing bilateral relationships, contribute to a more favorable market performance (Tavares 2009) and economic activity (Levine and Zervos 1998). However, during the sample period of the study, the behavior of the financial markets move according to monetary policies implemented by the Fed and European Central Bank, and deficit and debt structural adjustments in European economies. Thus, H8 is rejected for both Groups.

The results obtained from the current study evidence that green tech has a positive relationship with carbon productivity but no significance which may be justified, on the one hand, due to the low prices of energy-related fossil fuel (Samargandi 2017) and, on the other hand, due to the restrictions in patent applications concerning in technological diffusion and high costs associated (Mensah et al. 2018). Hence, H9 is rejected.

Although no hypothesis is formulated, the results show a positive relationship between green tech and aggregate material production in the context of Global Group, which appoints for as it ramps up $CO_2$ emissions higher probability that a country develops an environmental-related technology (Su and Moaniba 2017).

Despite the Russian crisis (caused by an emerging market) having a significant impact on emerging and developed financial markets (Dungey et al. 2006, 2007, 2010). However, in the present paper, the dummy Crises EM does not identify such evidence. It can be justified by the fact that emerging market crises are derived from exchange rate crises, and, therefore, the significant and positive effect of the Dummy Crises EM with greater evidence

in European countries can be justified through the appreciation of European currencies. In the global context, the effect is irrelevant because the Global Group includes emerging countries where financial crises were onset.

In turn, Dummy Crises DM, in the European context, has a negative and significant impact only on the stock market. This may indicate that there was contagion through the bond market channel (Dungey et al. 2010), which in this case, means sovereign debt and collateralized debt obligations markets. Furthermore, in the global context, it denotes a positive effect on economic activity, technology, and material production, which are associated with the aggressive monetary policies by FED and ECB (Dungey et al. 2006), smoothing the crises effect and contributing to the boost of the respective economies. Thus, it is determined that the crises are not alike, taking into account that the methodology applied in Dungey et al. (2006, 2007, 2010) is not at all similar to that of the present study. However, financial crises do not reveal a negative effect on energy efficiency and environmental degradation, which is contrasting with previous findings of Mimouni and Temimi (2018) and Pacca et al. (2020).

## 5. Conclusions

### 5.1. Empirical Findings and Implications

Considering the empirical findings, concerning the Global Group, material production is found to be the predominant factor in the positive determination of the behavior of FDI, economic activity and green tech. In this context, it is underlined that the industrialization process embarked on in recent decades, in emerging economies, has promoted the creation, acquisition and investment of companies in industrial sectors, contributing to global economic growth (positive relationship from aggregate material production to FDI and economic activity). Mergers and acquisitions among the largest listed metal and mineral companies can be connected to the positive effect on markets. In turn, the greater investment in new, cleaner and more efficient energy can be at the origin of increased carbon productivity (positive relationship from aggregate material production to green tech and positive relationship from economic activity to carbon productivity).

Regarding the EU-15 Group, the empirical findings reveal six significant relationships in which a shock from stock markets induces positive effects on aggregate material production and FDI; a shock from economic activity denotes a negative effect on aggregate material production; a shock from aggregate material production affects positively carbon productivity; a shock from FDI impact negatively carbon productivity; and a shock from carbon productivity denotes a positive effect on stock markets behavior. In addition, the stock market is the predominant factor.

It can be concluded that, in the European context, despite the endeavor by agents to reduce fuel energy dependence, the European economy is still based on a fuel energy model. However, there is a slight tendency to change the paradigm of a fuel energy model, which is observed based on negative relationship between economic activity and aggregate materials production, as well as through a positive relationship between aggregate material production and carbon productivity. An important factor of this scenario may be the launch of the European carbon market. Nevertheless, despite robust investment in R and D for achieving energy efficiency or carbon mitigation processes, it seems not to reach the continuous increase in $CO_2$ emissions (negative FDI and carbon productivity), which leads to the companies' financing moving forward to the green bond market as a driving force for green development and innovation (negative relationship between the stock market and FDI; positive relationship between carbon productivity and stock market) as well as for financing efficiency processes of production, resulting in a positive contagion toward to the stock market (positive relationship between markets and material production).

In short, the findings from the present empirical study verifies that material production is still a key driver of the global economy because production influences both macroeconomic fundamentals and innovation activities. For its turn, in the European economy context, material production only positively influences carbon productivity, which indi-



cates that this economic block seeks to be adopting an economic environmentally-friendly growth model. It should be noted that the dematerialization process is not detected in this study, in a global context, insofar as the economic activity keeps still based on an energy fuel model because fuel prices are considered lower comparing with the renewable energy prices. To ensure the transition into a complete green growth model, it is important to have a developed financial system that leads to strong incentives for green financing and energy efficiency, such as, for example, green, social and sustainable bond markets. No less important, it is worthy to emphasize the importance of emissions trading systems as important instruments to achieve a significant reduction in $CO_2$ emissions.

Overall, considering the dummy variables, it is concluded that recession in the most relevant economies leads to a mainly negative effect on the variables studied, whereas the short-term Kitchin cycles produce a mostly positive effect on the same variables. Therefore, it is verified that economies are more likely to react negatively to the economic recessions of the largest economies than to financial crises.

*5.2. Limitations and Future Research*

This study is not without limitations. Firstly, the period of the sample should be extended in future research, in order to increase the still limited knowledge about the global effects of the current pandemic crisis. Added to this is the limited number of countries analyzed, which needs enlarging in future studies. Secondly, there is the limitation of not being able to add other variables proxies of recycling activities and circular economy. This limitation is due to lack of access to available annual data that would allow us to expand this analysis for a larger set of materials.

In order to address the limitations mentioned above, future research could make an analysis to identify the response of the material supply and demand, in various performance regimes, in relation to the behavior of economic-financial variables, including macroeconomic fundamentals and indices of economies' digitalization and sophistication. Finally, it is also important to devote future research efforts to assessing the influence of recycling activities and of the circular economy on the behavior of carbonic productivity, taking into account the different states of economic activity.

**Author Contributions:** For conceptualization: J.L. and J.F.; methodology: J.L. and J.F.; redaction—original preparation of the draft: J.L. and J.F.; redaction—revision and edition: J.L. and J.F.; viewing: J.L. and J.F.; supervision: J.L. All authors have read and agreed to the published version of the manuscript.

**Funding:** This research was funded by the project EMaDeS—Energy, Material, and Sustainable Development EU/CCDRC/FEDER (Brussels/Coimbra, Central Region, Portugal) 2017 to 2021 | Central-01-0145-FEDER-000017.

**Institutional Review Board Statement:** Not applicable.

**Data Availability Statement:** The data can be made available from the authors upon request.

**Acknowledgments:** The authors acknowledge the highly valuable comments and suggestions provided by the editors and reviewers, which contributed to the improvement in the clarity, focus, contribution, and scientific soundness of the current empirical study.

**Conflicts of Interest:** The authors declare there are no conflict of interest. The funders had no role in the design of the study; in the collection, analyses, or interpretation of data; in the writing of the manuscript, or in the decision to publish the results.

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
