# Peer review of "Dynamic Effects of Material Production and Environmental Sustainability on Economic Vitality Indicators: A Panel VAR Approach"

_jrfm, doi:10.3390/jrfm14020074_

Round 1

Reviewer 1 Report

jrfm-1059346

Dynamic Effects of Material Production and Environmental Sustainability on Economic Vitality Indicators: A Panel VAR Approach

SUMMARY

This study analyses the relations and dynamics between material production, Foreign Direct Investment (FDI), economic activity, carbon productivity and the stock market, in a global and European context, using vector autoregressive methodology with panel data (PVAR). The empirical evidence obtained for the Global Group reveals four significant and positive one-directional relations, with FDI and material production being the dominant variables. For the EU-15 Group, nine significant relations were detected, among them one negative and two bi-directional, despite having opposite signs. Stock markets and FDI are found to be the most dominant variables, with FDI standing out as causing the greatest shock effect. Economic recessions show a generally negative effect, which contrasts with the economic cycles of Kitchin, which reveal a generally positive effect.

COMMENTS

1. Please check the dimensionalities in (1) as they're wrong.

2. If \mu_{i} is fixed effect why should it be correlated with Y_{i,t}?

3. The VAR model is based on no underlying economics.

Author Response

Dynamics effects of material production and environmental sustainability on economic vitality indicators: A panel VAR approach

Manuscript ID: jrfm-1059346

Dear Editor-in-Chief of the JRFM, Prof. Dr. Michael McAleer,

Firstly, we would like to thank all the reviewers for the constructive feedback and suggestions concerning the previous version of the manuscript. Secondly, we are very pleased to have had the opportunity to revise and resubmit the paper. Considering the responses to the questions raised, we provide a global overview of what was changed according to the review proposals and constructive suggestions made by the reviewers.

Yours faithfully

The Authors

Reviewer 1 (Rev_1.)

Comment No.

Page

No.

Section

Reviewer 1: Comments

Amendments

Rev_1_1

5

3.1. Econometric model

Please check the dimensionalities in (1) as they're wrong.

We acknowledge the reviewer’s comment and constructive feedback. Considering the comment, we added in the text the following corrected expression: i ∈ {1,2,…,N}, t ∈ {1,2,…,Ti}

where: i corresponds to countries encompassed in the present study; t is the time horizon for each i;  is a kx1 vector of endogenous variables;   is a lx1 vector of exogenous variables;  is a 1xk vector of individual fixed effects; and and  is a 1xk  vector of idiosyncratic errors ( ~ i.i.d.).The kxk matrices: , ,…,, ; and the lxk matrix: ; represent the estimated parameters. Therefore, the PVAR model assumes that cross-sections hold same units in data generating process, which result in common parameters in matrixes: , ,…,, ; and ; encompassing heterogeneity through panel-specific fixed effects (Holtz-Eakin et al. ,1988; Abrigo and Love, 2016).

Rev_1_2

5

3.1. Econometric model

If \mu_{i} is fixed effect why should it be correlated with Y_{i,t}?

Considering the reviewer’s comment, we would like to clarify that the mu_{i} is correlated with regressors inducted by considering the lagged dependent variables. In case of use of the OLS and standard mean-differencing method, this could cause biased model estimates. For addressing this constraint, it is used the Helmert transformation, which removes the mean of future observations, applying the GMM estimator that uses lagged covariates as instrumental variables (Arellano and Bover, 1995).

Rev_1_3

2

1.Introduction

The VAR model is based on no underlying economics.

Taking into consideration the reviewer’s comment, we would like to justify the option for using the selected order of variables’ entry, which uses the Cholesky decomposition, considering the exogeneity level of each variable, as justified in lines 234 and 235 (i.e. after the equation (3)). Therefore, in the introduction section is now argued that the PVAR model specification is not based on prior Theory, adding the following sentence:

In this line of analysis, this study aims to analyze the behavioral effects and dynamics of shocks between material production, environmental sustainability and economic-financial variables, using the panel VAR (PVAR) methodology (Love and Zicchino, 2006; Abrigo and Love, 2016). Following the same previously referred authors, this advanced econometric methodology prevents endogeneity issues. In addition, it is not based on prior theory regarding variables’ relationships, and provides the possibility of using two forecasting techniques, such as the orthogonalized impulse-response functions, and the forecast error decomposition variance, to gauge forecast effects on the system. Comparing to the VAR model, the current methodology in use, allows heterogeneity in panel estimation procedures.

Reviewer 2 Report

This paper provides a panel VAR analysis of the inter-relationships between materials production, FDI, economic activity, carbon emissions and the stock market. The paper also provides evidence on the presence of the Environmental Kuznets Curve (EKC).  The panel VAR analysis also involves presentation of forecast error variance decompositions and impulse response functions.

The paper could be improved by consideration of the following point:

A crisis dummy variable is created with a value of 1 in crisis years and zero otherwise, for details see footnote 4. This treats all the crises as being the same - is this a reasonable approach or could you distinguish between the crises that have commenced in developed markets from those that commenced in emerging markets? This goes the question of are all crises the same? At a general level you could consider the discussion in Dungey at al (2006, 2007, 2010) on this matter, and also the literature that has explored crisis impacts of energy and environmental matters, see for example Mimouni and Temimi (2018) and Pacca et al (2020). Perhaps add some discussion of this literature in the discussion of your crisis results, and also consider a robustness check that treats some of the crises differently.

References

M Dungey, R Fry, B González-Hermosillo, V L Martin (2006), Contagion in International Bond Markets During the Russian and LTCM Crises, Journal of Financial Stability, 2, p. 1 – 27

M Dungey, R Fry, B González-Hermosillo, V L Martin (2007), Contagion in Global Equity Markets in 1998: The Effects of the Russian and LTCM Crises, North American Journal of Economics and Finance, 18, p. 155 - 174

M Dungey, R Fry, V L Martin, C Tang, B González-Hermosillo, (2010), Are Financial Crises Alike? IMF Working Paper No. 10/14, Available at SSRN: https://ssrn.com/abstract=1537510

K Mimouni, A Temimi, (2018) What drives energy efficiency? New evidence from financial crises, Energy Policy, 122, p. 332-348

L Pacca, A Antonarakis, P Schroder, A Antoniades, (2020), The effect of financial crises on air pollutant emissions: An assessment of the short vs. medium-term effects, Science of the Total Environment, 698, art. no. 133614

Author Response

Dynamics effects of material production and environmental sustainability on economic vitality indicators: A panel VAR approach

Manuscript ID: jrfm-1059346

Dear Editor-in-Chief of the JRFM, Prof. Dr. Michael McAleer,

Firstly, we would like to thank all the reviewers for the constructive feedback and suggestions concerning the previous version of the manuscript. Secondly, we are very pleased to have had the opportunity to revise and resubmit the paper. Considering the responses to the questions raised, we provide a global overview of what was changed according to the review proposals and constructive suggestions made by the reviewers.

Yours faithfully

The Authors

Reviewer 2 (Rev._2)

Comment No.

Page

No.

Section

Reviewer 2: Comments

Amendments

Rev.2_1

3.2 Data and specification of the model

A crisis dummy variable is created with a value of 1 in crisis years and zero otherwise, for details see footnote 4. This treats all the crises as being the same - is this a reasonable approach or could you distinguish between the crises that have commenced in developed markets from those that commenced in emerging markets?

We acknowledge the reviewer’s comment and constructive feedback. Considering the comment and valuable suggestion, the crises dummy was divided in two dummies: (1) the Dummy Crises EM, where value 1 depicts the crises that were originated in emerging markets, and 0, otherwise; and (2) the Dummy Crises DM, where value 1 depicts the crises that have commenced in developed markets and 0, otherwise. Please, see the explanation provided in the footnote 4.

The dummy variable D^(Crises EM) has the value of 1 in the annual periods of 1991, 1994, 1995, 1997-2000, 2002, and the value of zero in the remaining periods. The periods under analysis correspond to different international crises, such as: the oil crisis (1991); the Mexican economic crisis (1994/1995); the Asian monetary crisis (1997); the Russian monetary crisis (1998); the Brazilian monetary crisis (1999); the Argentinian economic crisis (1999-2000); and the South American economic crisis (2002). The dummy variable D^(Crises DM) equals to 1 in the annual periods of 2001 and 2007-2010, and 0 in other periods .These periods correspond to the dotcom bubble (2001), the subprime crisis (2007-2008) and the European debt crisis (2009-2010).

Rev.2_2

4.3 Discussion

This goes the question of are all crises the same? At a general level you could consider the discussion in Dungey at al (2006, 2007, 2010) on this matter, and also the literature that has explored crisis impacts of energy and environmental matters, see for example Mimouni and Temimi (2018) and Pacca et al (2020). Perhaps add some discussion of this literature in the discussion of your crisis results, and also consider a robustness check that treats some of the crises differently.

Considering the reviewer’s comment, the reference discussions provided by Dungey et al. (2006, 2007, 2010), and the previous findings of Mimouni and Temimi (2018), and Pacca et al. (2020), were incorporated for guiding the discussion of the empirical findings now obtained, contrasting the findings with previous results, adding the following paragraphs:

Despite the Russian crisis (caused by an emerging market) having a significant impact on emerging and developed financial markets (Dungey et al.2006; 2007; 2010). However, in the present paper, the dummy Crises EM does not identify such evidence. It can be justified by the fact that emerging market crises are derived from exchange rate crises and, therefore, the significant and positive effect of the Dummy Crises EM with greater evidence in European countries, can be justified through the appreciation of European currencies. In the global context, the effect is neither relevant, since the Global Group includes emerging countries where financial crises were onset.

In turn, Dummy Crises DM, in the European context, has a negative and significant impact only on the stock market. This may indicate that there was contagion through the bond market channel (Dungey et al., 2010), in this case, means sovereign debt and collateralized debt obligations markets. Furthermore, in the global context, it denotes a positive effect on economic activity, technology, and material production, which are associated with the aggressive monetary policies by FED and ECB (Dungey et al. 2006), smoothing the crises effect and contributing to the boost of the respective economies. Thus, it is determined that the crises are not alike, taking into account that the methodology applied in Dungey et al. (2006; 2007; 2010) is not at all similar to that of the present study. However, financial crises do not reveal a negative effect on energy efficiency and environmental degradation, which is contrasting with previous findings of Mimou and Temimi (2018) and Pacca et al. (2020).

Reviewer 3 Report

Please refer to the attached report.

Author Response

Dynamics effects of material production and environmental sustainability on economic vitality indicators: A panel VAR approach

Manuscript ID: jrfm-1059346

Dear Editor-in-Chief of the JRFM, Prof. Dr. Michael McAleer,

Firstly, we would like to thank all the reviewers for the constructive feedback and suggestions concerning the previous version of the manuscript. Secondly, we are very pleased to have had the opportunity to revise and resubmit the paper. Considering the responses to the questions raised, we provide a global overview of what was changed according to the review proposals and constructive suggestions made by the reviewers.

Yours faithfully

The Authors

Reviewer 3 (Rev. 3)

Comment No.

Page

No.

Section

Reviewer 3: Comments

Amendments

Rev.3_1

1. Introduction

The question of sustainable economic growth is relevant and worth investigation. However, in my opinion, the focus and the research contributions of this paper are not very clear.

We acknowledge the reviewer’s comment and constructive feedback. Following the suggestion, in the introductory item the contributions into the literature of sustainable finance are provided in the following sentence:

The present empirical study presents a two-fold contribution for the literature on sustainable finance: (1) analyzing the still unexplored relationships between materials production, green growth, innovativeness, and macroeconomic fundamentals, in order to deepen the knowledge on how to foster the production and financial strategies oriented to a green economy pathway; and (2) unveiling dematerialization paths, in terms of the relationship between sustainable/green growth and macroeconomic fundamentals, considering the short cycles of Kitchin.

Rev.3_2

5.1

Empirical findings and implications

Although the Introduction focuses on dematerialization and sustainable economic growth, it is not clear how the paper

contributes to research and literature about these topics.

Bearing in mind the reviewer’s comment, the contributions were outlined both in the introductory item (please see answer to the previous comment) an in the conclusions item, as stated below:

In short, the findings from the present empirical study ratify that material production is still a key driver of the global economy since it influences both macroeconomic fundamentals and innovation activities. For its turn, in the European economy context, material production only positively influences carbon productivity, which indicates that this economic block seeks to be adopting an economic environmental-friendly growth model. It should be noted that the dematerialization process is not detected in this study, in a global context, insofar as the economic activity keeps still based on an energy fuel model, since that fuel prices are considered lower comparing with the renewable energy prices. To ensure the transition into a complete green growth model, it is important to have a developed financial system that leads to strong incentives for green financing and energy efficiency, such as, for example, green, social and sustainable bond markets. No less important, it is worthy to emphasize the importance of Emissions Trading Systems as important instruments to achieve a significant reduction in greenhouse emissions.

Rev.3_3

2-5

2.Theoretical framework, evidence and hypotheses

The eight research questions stated in Section 2 are not formulated as truly researchable hypotheses. The authors should state explicitly what they mean by \relationship" (contemporary, lagged, reciprocal,Granger-causality...)

We acknowledge the reviewer’s comment and constructive feedback. Considering the comment, the research hypotheses were revised, according to the sense of Granger-causality relationship:

H1: The economic activity and material production denote a negative causality relationship.

H2: The economic activity and carbon productivity present a positive causality relationship.

H3: The stock market and material production denote a negative causality relationship.

H4: The stock market and carbon productivity have a positive causality relationship.

H5: The FDI and carbon productivity have a negative causality relationship.

H6: The FDI and stock market denote a positive causality relationship.

H7: The FDI and economic activity present a positive causality relationship.

H8: The stock market and economic activity have a positive causality relationship.

More recently, the technological innovation appears in the global policy agenda as a means of carbon mitigation and for the transition to a sustainable and green economy. However, although some empirical evidence shows that technological innovation becomes an important factor in carbon mitigation (Fernández et al. 2018), other studies identify the possibility of a rebound effect (Magee and Devezas 2017; Wang et al., 2019; Cheng et al. 2019) or from another angle, technological development does not decrease gas emission (Samargandi 2017; Mensah et al. 2018). Considering the previous evidences, the following hypothesis is considered:

H9: The carbon productivity and green technology have a negative causality relationship.

Rev.3_4

2-5

3.2. Data and specification of the model

Which macroeconomic variables they used to specify some broad-based concepts (e.g., economic activity, stock market, material production).

We acknowledge the reviewer’s comment and constructive feedback. In this sense, a new Table (e.g. Table 1), presenting the variables, associated concepts, description, units, and statistical sources, was introduced in the sub-section: 3.2 Data, variables, and specification of the model; of the revised version of the manuscript.

Rev.3_5

5-6

1.Introduction

The choice of a PVAR model should also be motivated, possibly in comparison with alternative econometric models.

Considering the reviewer’s comment, in the introductory item, a justification for using a PVAR model is provided in the following sentence:

In this line of analysis, this study aims to analyse the behavioural effects and dynamics of shocks between material production, environmental sustainability and economic-financial variables, using the panel VAR (PVAR) methodology (Love and Zicchino, 2006; Abrigo and Love, 2016). Following the same previously referred authors, this advanced econometric methodology prevents endogeneity issues. In addition, it is not based on prior theory regarding variables’ relationships, and provides the possibility of using two forecasting techniques, such as the orthogonalized impulse-response functions, and the forecast error decomposition variance, to gauge forecast effects on the system. Comparing to the VAR model, the current methodology in use, allows heterogeneity in panel estimation procedures.

Rev.3_6

6-7

3.2. Data and specification of the model

The dataset covers 24 countries for a period of 27 years. In these circumstances, an unrestricted PVAR model could be over parameterized and \the number of potential restricted PVAR models of interest could be huge" (see, e.g., Koop and Korobilis, European Economic Review, 2016).

Unfortunately, the model specification (1) given in the paper is not

followed by appropriate information about the parameterization adopted. In particular, the coefficients A1,…,Ap seem to be the same for all countries i =1,…,N, a strong assumption that is not discussed in the paper.

We acknowledge the reviewer’s comment and constructive feedback. Considering the comment, the following changes and amendments were mad in the item 3.1.:

The kxk matrices: , ,…,, ; and the lxk matrix: ; represent the estimated parameters. Therefore, the PVAR model assumes that cross-sections hold same units in data generating process, which result in common parameters in matrixes: , ,…,, ; and ; encompassing heterogeneity through panel-specific fixed effects (Holtz-Eakin et al. ,1988; Abrigo and Love, 2016).

Rev.3_7

3.2. Data and specification of the model

Moreover, the use of dummy variables for global and country-specific crises is, in my opinion, questionable. Indeed, these crises are an intrinsic part of the economic activity of each country, and should not be reduced to exogenous dummy variables.

We acknowledge the reviewer’s comment and constructive feedback. Considering the comment, which is aligned with the first suggestion provided by Reviewer 2, the crises dummy was divided in two dummies: (1) the Dummy Crises EM, where value 1 depicts the crises that were originated in emerging markets, and 0, otherwise; and (2) the Dummy Crises DM, where value 1 depicts the crises that have commenced in developed markets and 0, otherwise. Please, see the explanation provided in the footnote 4.

The dummy variable D^(Crises EM) has the value of 1 in the annual periods of 1991, 1994, 1995, 1997-2000, 2002, and the value of zero in the remaining periods. The periods under analysis correspond to different international crises, such as: the oil crisis (1991); the Mexican economic crisis (1994/1995); the Asian monetary crisis (1997); the Russian monetary crisis (1998); the Brazilian monetary crisis (1999); the Argentinian economic crisis (1999-2000); and the South American economic crisis (2002). The dummy variable D^(Crises DM) equals to 1 in the annual periods of 2001 and 2007-2010, and 0 in other periods .These periods correspond to the dotcom bubble (2001), the subprime crisis (2007-2008) and the European debt crisis (2009-2010).

Rev.3_8

3.2. Data and specification of the model

The concepts of economic activity, material production and environmental sustainability could be related to many observable variables which, in their turn, are strongly connected to the population dynamics of each country. The paper should include a more detailed discussion about the proxy variables chosen by the authors and specify whether per-capita indicators were used or not.

We acknowledge the reviewer’s comment and constructive feedback. Thus, we clarify the proxies and units used in Table 1, as previously answered in comment 4 of the Reviewer 3.

Rev.3_9

5.2. Limitations and future research

 The proposed model should be possibly extended with the inclusion of additional variables representing, for instance, recycling activities, circular economies, new technologies.

We acknowledge the reviewer’s comment and valuable suggestion. To address this, a new explanatory variable was introduced in the model specification, that is, environmental technologies (ENV_TECH), and the new PVAR model was estimated, both for the Global panel and the EU-15 panel. Concerning recycling activities and circular economies, it was not possible, so far, to have access to available annual data, but this is outlined as a future research avenue, in the conclusions item, including the following sentence:

Finally, it also important to devote future research efforts for assessing the influence of recycling activities and of the circular economy, on the behavior of carbonic productivity, taking into account the different states of economic activity.

Round 2

Reviewer 1 Report

I am happy with the revised version.

Reviewer 2 Report

The authors have addressed my comments on the previous version and I would now be happy to recommend acceptance.

Reviewer 3 Report

The Authors have addressed all my comments and have revised the paper accurately.